# Aligner-Encoders:
# Self-Attention Transformers Can Be Self-Transducers

**Adam Stooke**
Google, USA
astooke@google.com

**Rohit Prabhavalkar**
Google, USA

**Khe Chai Sim**
Google, USA

**Pedro Moreno Mengibar**[*]

## Abstract

Modern systems for automatic speech recognition, including the RNN-Transducer and Attention-based Encoder-Decoder (AED), are designed so that the encoder is not required to alter the time-position of information from the audio sequence into the embedding; alignment to the final text output is processed during decoding. We discover that the transformer-based encoder adopted in recent years is actually capable of performing the alignment internally during the forward pass, prior to decoding. This new phenomenon enables a simpler and more efficient model, the "Aligner-Encoder". To train it, we discard the dynamic programming of RNN-T in favor of the frame-wise cross-entropy loss of AED, while the decoder employs the lighter text-only recurrence of RNN-T without learned cross-attention—it simply scans embedding frames in order from the beginning, producing one token each until predicting the end-of-message. We conduct experiments demonstrating performance remarkably close to the state of the art, including a special inference configuration enabling long-form recognition. In a representative comparison, we measure the total inference time for our model to be 2x faster than RNN-T and 16x faster than AED. Lastly, we find that the audio-text alignment is clearly visible in the self-attention weights of a certain layer, which could be said to perform "self-transduction".

## 1 Introduction

The task of sequence transduction requires mapping from an input sequence to an output sequence. It appears in several widespread applications including machine translation and automatic speech recognition (ASR). In ASR, which is the focus of this paper, the two sequences lie in completely different modalities, and the audio input representation is typically much longer than the output text. Thus, in addition to identifying and converting speech sounds, the model must find an "alignment" that moves information from wherever it appears in the input sequence to wherever it belongs in the output. Among other issues, the many possible variations in the pace of pronunciation make transduction a challenging problem.

More than a decade ago, and almost a decade apart, two powerful algorithms relying on dynamic programming were developed to perform the sequence transduction task with recurrent neural networks (RNNs). The motivation behind these algorithms was that RNNs require training every output frame in the sequence—with one output per input frame, they needed to be trained with alignments already prepared. The new algorithms allowed training without prepared alignments by computing the probability of the label marginalized over *all possible alignments* as the optimization objective. The differences in sequence length are accommodated by learning to output "blank" symbols in between true labels and post-processing them out at inference time. The first algorithm, Connectionist Temporal Classification (CTC) [1], models the output frames as conditionally independent, and this

---

[*]Work performed while at Google, USA.

38th Conference on Neural Information Processing Systems (NeurIPS 2024).

limitation was overcome in the follow-on work RNN-Transducer (RNN-T) [2], which introduced auto-regressive decoding to achieve better performance.

A third algorithm appeared shortly after, called attention-based encoder-decoder (AED). It uses a learned mechanism to account for alignments [3–6]; the encoder is followed by a recurrent decoder that cross-attends at every step between its RNN state and the entire encoder embedding sequence to produce its next state. The cross-entropy loss is applied straightforwardly between the plain label sequence and the leading output frames, with no blank symbol and no dynamic programming required. The resulting model produces its output auto-regressively until an end-of-sentence (<EOS>) token halts decoding. AED models, together with CTC and RNN-T, have propelled end-to-end deep neural networks to become the best performing ASR systems across academia and industry (*e.g.*, see surveys [7, 8]).

Despite their great successes, each of these algorithms has its downsides. Implementing the probability summations for CTC and especially RNN-T require non-trivial effort. A naive formulation of the principle is intractable to compute, requiring a dynamic programming approach, which in turn has been the subject of optimization efforts by expert practitioners [9–12]. Beyond that, RNN-T training requires computing potentially unwieldy tensor quantities to do with cross-pairing every frame of input with every token of output. Both models suffer inefficiencies in decoding. RNN-T processes all frames recurrently while producing a typically much shorter output sequence. AED requires computing over the entire encoder embedding sequence at every decoding step, which leads to slow operation from the high compute load within the auto-regressive loop.

In an effort to resolve these difficulties, we ask the question: with the advent of transformer-based encoders [13–20], can the ASR *encoder* itself learn to perform the alignment? The answer we found is: yes, it can. Our main contribution is to show that it is now possible to train neural network speech recognizers with light-weight decoders in the style of RNN-T, but with the simple frame-wise cross-entropy loss of AED. The resulting networks provide accuracy remarkably close to state-of-the-art models while being more efficient at training and decoding than any previous model. No explicit dynamic programming is employed; instead, the encoder learns to perform the alignment internally during the forward pass. The encoded embedding frames are decoded consecutively from the beginning, one at a time, in conjunction with a small recurrent network that only reads in the previous label, producing exactly one label per frame until emitting <EOS>.

Having originated in the era of long short-term memory (LSTM) [21, 22] encoders, the preceeding algorithms all allow the encoder to process information in-place in the time dimension, resulting in embeddings with relevant information in roughly the same position in the sequence as it appeared in the input. For several years now, however, ASR systems have benefited from the adoption of more powerful transformer encoders, but without changing their role. Our Aligner-Encoders are different in that, simultaneous to encoding, they perform the additional task of moving the relevant information to the beginning of the embedding sequence into a label-aligned position. As it is performed solely through the self-attention mechanism within the forward pass of the encoder, one might call this a "self-transducer". The previous style of information flow is contrasted with ours in Figure 1. A main benefit of our model utilizing an Aligner-Encoder is that it is dramatically simpler to conceptualize, implement, and use.

This paper is organized as follows. First, we describe Aligner-Encoder models and relate them to RNN-T and AED. Next, we relate other works which have aimed at improving RNN-T modeling effectiveness or efficiency. Then we report experiments demonstrating the accuracy of Aligner models while also finding limitations for generalizing to long test utterances. In response, we introduce techniques which can be employed at inference-time to provide good long-form performance without any additional training. In further experiments, we analyze how and when Aligner-Encoders perform the alignment and make the surprising discovery that it can happen primarily within a single self-attention layer. Lastly, since ASR alignments are monotonic, we demonstrate that Aligners can at least readily handle *reverse* alignments, making our model promising for future work in non-monotonic applications such as machine-translation or speech-translation.

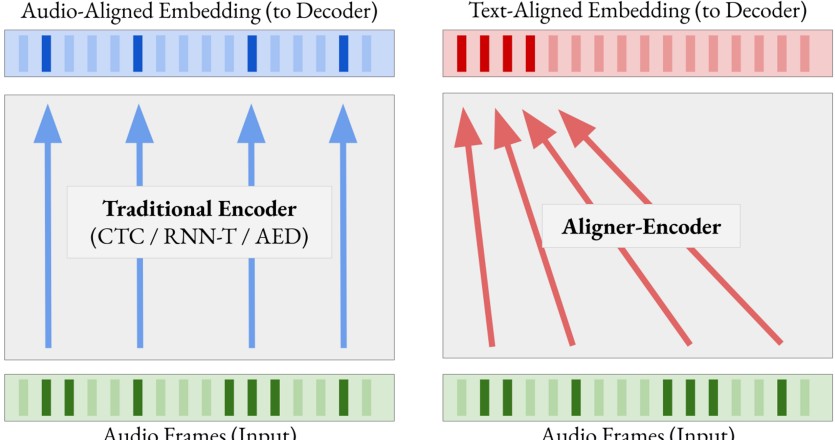

Figure 1: Information flow through an Aligner-Encoder versus traditional audio encoders.

## 2   Model

### 2.1   Aligner-Encoder Model

The Aligner-Encoder model combines the best elements of RNN-T and AED systems into a more compact form. By requiring the encoder itself to text-align its embedding, we avoid using 1) dynamic programming to sum probabilities in the loss and 2) full-sequence cross-attention in the decoder (in all models we refer to everything inside the auto-regressive portion as the "decoder"). It may be simplest to first lay out our model, which is formulated using the same components as RNN-T, and afterwards draw contrasting points with the heritage.

We begin with an input sequence $\mathbf{x} = (x_1, x_2, ..., x_T)$ of length $T$, and output sequence $\mathbf{y} = (y_1, y_2, ..., y_U)$ of length $U \leq T$. An *encoder* network, $f_{\text{enc}}$, processes the input sequence into an acoustic embedding sequence, $\mathbf{h} = (h_1, h_2, ..., h_{T'})$, where each frame of the embedding sequence can depend on every frame of input, and typically $T' \leq T$ with subsampling. A *prediction network*, $f_{\text{pred}}$, processes text labels with forward recurrence only to produce a text embedding sequence, $\mathbf{g} = (g_1, g_2, ..., g_U)$. The acoustic and text embeddings are fed into a *joint network*, $f_{\text{joint}}$, to produce the final prediction vector of dimension $V$ according to the vocabulary, for each frame, with softmax probability normalization. The difference in our model is that we enforce the alignment *at the encoder output* by restricting the model to use the acoustic and text embedding frames in a one-to-one fashion (Equation (3)). The entire model is written by its recurrence relations as:

$$\mathbf{h} = f_{\text{enc}}(\mathbf{x}) \tag{1}$$

$$g_i = f_{\text{pred}}(g_{i-1}, y_{i-1}), \quad i \leq U \tag{2}$$

$$P(y_i | \mathbf{x}, y_{<i}) = f_{\text{joint}}(h_i, g_i), \quad i \leq U \tag{3}$$

The encoder and the decoder–which includes both the prediction and joint networks–are parameterized and learned together in an end-to-end manner, with total parameters $\theta$. We maximize the log probabilities of the correct labels, resulting in the familiar cross-entropy loss:

$$\mathcal{L}_{Aligner}(\theta) = -\sum_{i=1}^{U} \log P(y_i | \mathbf{x}, y_{<i}; \theta) \tag{4}$$

The loss only applies to encoder frames within the length of the label, $T' \leq U$; all remaining frames ($T' > U$) are ignored. Hence the *encoder* must also learn to be an *aligner*.

We seed the prediction network with a start-of-sentence token, `<SOS>`, at the first step, $y_0$, and empty state $h_0 = 0$. As is often the case with AED and other uses of cross-entropy loss, we find label smoothing to be a beneficial regularizer and always use it [23–25] (weight $0.1$). During inference the label length is unknown, so the model must predict it, as in AED. We train the model to always predict the end-of-sentence token, `<EOS>`, as the final token of every training example[2], so decoding proceeds token-by-token until `<EOS>` is predicted.

---

[2]We include the `<EOS>` in the label count $U$, without loss of generality.

It is also possible to formulate a non-autoregressive (non-AR) Aligner, which applies a decoder, $f_{\text{ind}}$, to the embedding frames independently, as in CTC:

$$P(y_i|\mathbf{x}) = f_{\text{ind}}(h_i) \tag{5}$$

and it can be trained under the same loss. During inference, all tokens are predicted independently, and any tokens after the earliest `<EOS>` in the sequence are discarded. We include an experiment with this formulation; however, we found its performance to be significantly worse than CTC in all but the shortest-utterance datasets.

## 2.2 Advantages over RNN-T & AED

Given that previous models were developed to overcome contemporary neural encoders' inability to align, the new model naturally brings simplifications. RNN-T trains by explicitly marginalizing over all possible alignments in the loss. This can be visualized in a label-frame decoding lattice (see Figure 1 in [2]), where each node represents an {encoder-frame, text-frame} pair, and every pairing is a valid state to be considered, resulting in a $U \times T$ rectangular grid (abbreviating $T'$ as $T$). Beyond requiring a sophisticated dynamic programming implementation to calculate tractably (involving forward and backward probabilities scanned across the lattice), the marginalization still leads to computing all $U \times T \times V$ logits of the lattice, a potentially memory-intensive step (it is computing Equation (3) over all pairs of indices as $f_{\text{joint}}(h_i, g_j)$). The Aligner loss can be viewed as prescribing only the main diagonal of the lattice to be valid, reducing the realized logits to the bare minimum $U \times V$.

Savings come during inference, as well. The decoder in AED operates with $O(U \times T)$ complexity, since it cross-attends to the entire encoder embedding sequence at every step, for $U$ steps. Furthermore the constant factor is often relatively large in practice since a more powerful decoder network is needed for computing the attention (the recurrence relation is $g_i = f_{\text{dec,AED}}(\mathbf{h}, g_{i-1}, y_{i-1})$, compare at our Equation (2) which lacks $\mathbf{h}$ or even $h_i$). The RNN-T decoder has complexity $O(T + U)$, since it must emit a blank at every frame to advance to the end of the lattice in a addition to the steps emitting a label. In contrast, by preparing the alignment within the encoder, our model reduces decoder complexity to $O(U)$, the most efficient possible for auto-regressive token prediction. In practice our constant factor will also be small like RNN-T.

One final savings relative to RNN-T comes when using beam search during inference, which can significantly improve overall accuracy. The emissions of blank tokens in RNN-T means that the same text hypothesis can be represented by different paths through the decoding lattice. Thus a proper search requires a routine to combine the probabilities from equivalent lattice paths at every step. Known as "path merging", it can actually be an expensive operation, leading others to sometimes approximate the search without it, despite the potential loss of beam diversity (*e.g.*, see discussion in [26]). Since our model, like AED, does not emit any blank tokens, path merging does not apply.

## 3 Related Works

Before end-to-end ASR, the previous generation of "hybrid" ASR systems [27] relied on a separate base system to provide frame-level alignments which could then be used to train frame-level neural network acoustic models. Most modern ASR systems use an end-to-end framework; the most prominent modeling techniques are CTC [1], RNN-T [2], and AED [4, 5].

A number of works have aimed at modifying the base RNN-T model in order to gain efficiency or efficacy. In Monotonic RNN-T [28], the loss and decoder are modified to step forward every frame, preventing multiple token emissions per label and speeding up the decoder complexity to $O(T)$. More recently, [12] proposed a method to reduce memory usage of the RNN-T loss, especially for cases with large vocabulary size such as Chinese character-based, by using a linearized joint network first to identify a relevant sub-region of the lattice likely to contain the true alignment. They only compute the full joint network on that sub-region and still train successfully. Another issue with RNN-T systems arises when trying to treat the prediction network as a language model, which can be improved through text-only training, because its language modeling function is polluted by having to predict blanks. To ameliorate this, multiple works [29–31] have introduced a separate network component for predicting blanks, and re-formulated the overall prediction model accordingly, resulting in much greater receptiveness to language training. One aim of our design was to avoid the use of blank tokens entirely, although in the present work we do not pursue additional text training.

Monotonic alignments have also been developed for attention-based systems [32], requiring newly devised loss functions to enforce monotonicity differentiably and encourage discreteness. Integrate-and-fire systems are another approach proposed to limit the context window needed for cross-attention to the encoder embedding, by introducing a soft monotonic alignment mechanism [33–35]. It steps forwards through the encoder frames one at a time, accumulating weighted information from the embedding until an activation threshold is reached, upon which a token is emitted. CTC and attention-based systems have previously been combined [36, 37] resulting in improved learning and decoding, and it is not uncommon to train AED models with a CTC auxiliary loss. We did not pursue the analogous combination of non-AR and AR decoders for Aligners, although it would be simple to implement. Another interesting, concurrent line of work is the Bayes-CTC [38] and Bayes-Transducer [39] models, which add a modulating factor to the respective losses in order to encourage earlier token emission in decoding; however, only our simpler model achieves full text alignment with no blank embedding frames between tokens.

A common way to improve the efficiency of ASR systems is to downsample the time dimension within the encoder, known as frame reduction (*i.e.*, $T' < T$; [40]). In addition to reducing encoder complexity, which scales as $O(T^2)$ with self-attention, frame reduction yields decoder efficiencies. It was necessary in the original Listen, Attend Spell (LAS) model [5] to enable learnability for the cross-attention by presenting a manageable number of frames to the decoder. In RNN-T it reduces the number of decoder steps, which scale with $T$, and can in some circumstances be carried out to an extreme, packing multiple labels per frame [41]. In the few cases we tried, we found Aligners to have similar robustness to downsampling as RNN-T, but we did not study this in-depth. Aligners are like CTC in that they cannot downsample the encoder to fewer frames than the length of the text sequence, although perhaps they could be trained to decode multiple tokens per frame.

## 4   Experiments

We conducted a range of experiments to explore the performance of Aligners and compare them against previous models on an equal footing. After describing the common configurations and datasets used, we present three different sets of experiments. The first explores the performance of basic Aligner models, the second studies techniques to employ at test-time to attain long-form recognition, and the final set examines the alignment process occurring within the encoders.

### 4.1   Datasets

We experiment on three U.S. English datasets with very different characteristics. The first is LibriSpeech-960 hour (LS) [42]. The second is a Voice Search (VS) dataset comprised of real traffic, totalling over 100M utterances and nearly 500k hours of speech, with an average duration of 3.4 seconds; utterances are anonymized before processing. The majority of the utterances are pseudo-labeled by a teacher model [43], and a small portion are human transcribed. Only 5% of the queries are greater than 7.6 seconds long. The test sets for VS include a main set which covers many common types of utterances, and several rare-word sets generated using TTS from specific domains–maps, queries, and news. Lastly, we include a long-form dataset drawn from random YouTube videos (YT). The training set is comprised of utterances with pseudo-labels generated by a previous ASR system [43], cleaned to ensure no speaker overlap occurs, and ranging from 5-15 seconds each. The total training set size is roughly 670K hours. The test set includes 30 hours of long-form audio, at an average of 8 minutes per example, spanning a range of topic sources including lifestyle, entertainment, sports, gaming, and society. In all datasets, the audio input is represented using log-mel features with a 32ms window size at a 10ms stride.

### 4.2   Neural Networks Specifications

We used the same neural encoder architecture for every algorithm, with settings adjusted for each dataset as listed in Table 1. Our networks begin with learnable 2-D convolutional subsampling layers, with kernel size 3 and stride 2, 128 features in the first layer and 32 thereafter. We employ state-of-the-art Conformer encoders [20], which include in each layer: a feed-forward unit, a multi-headed self-attention layer, a 1-D convolution layer for localized processing, followed by an outgoing feed-forward unit and finally residual connection. The number of layers and model dimension vary by dataset, as shown in the table. We did not find RNN-T nor Aligner experiments to be highly sensitive

to decoder dimensions–they can operate with surprisingly small prediction networks–although results sometimes did improve slightly with larger prediction networks when also using a wider beam search.

All our models use a word-piece tokenizer [44], as is common in state-of-the-art systems. Word pieces may be as small as a single letter, or could be an entire word, such as "wednesday". Unlike phonemes, word pieces of a given vocabulary exhibit a wide range in duration, not to mention complexity, of audio content associated with them. Although this likely makes the alignment problem more difficult, word piece vocabularies are very effective because they cover the text distribution efficiently. For label smoothing, we estimate the word-piece prior on-the-fly using batch-wide label counts.

Table 1: Settings used with each dataset: LibriSpeech, Voice Search, and YouTube.

| Setting | LS | VS | YT |
|---|---|---|---|
| Log-Mel Features | 80 | 128 | 128 |
| 2-D Conv Layers | 2 | 2 | 3 |
| Enc Dimension | 512 | 768 | 1024 |
| Enc # Layers | 17 | 24 | 24 |
| Enc # Params | 100M | 300M | 600M |
| LSTM Size | 1x640 | 1x256 | 1x1024 |
| Vocab Size | 1024 | 4096 | 4096 |

To enhance the beam search, we apply label smoothing debiasing [45], which essentially removes low-probability tokens from the posterior and then re-normalizes it. Similar to [45], we find that a relatively large debiasing parameter of 2 (so the threshold becomes $2/V$) is helpful, and we use it with a modest beam size of 6 in our experiments for a small but consistent improvement in accuracy.

## 4.3 Base Model Results

Table 2 shows results in Voice Search, including the rare-word test sets, and Aligners perform comparably to RNN-T in all cases. We also compare CTC and non-AR Aligners, which perform nearly as well in Voice Search. We found that increasing the vocabulary size to 32K improved the performance of the non-AR Aligner, yet it still suffers from many deletions on the NEWS rare-word set, which contains longer utterances. Our non-AR model performed relatively poorly on LibriSpeech and YouTube. For sequences

Table 2: WER (%) on the Voice Search test sets. VS: Main Test, and rare-words RM: Maps, RN: News, RQ: Search Queries

| | VS | RM | RN | RQ |
|---|---|---|---|---|
| RNN-T | 3.6 | 12.6 | 14.6 | 20.5 |
| ALIGNER | 3.7 | 12.5 | 13.1 | 20.4 |
| CTC | 4.3 | 14.3 | 17.8 | 23.8 |
| NON-AR ALIGNER | 4.5 | 15.3 | 27.3 | 23.5 |

beyond a very short length, an auto-regressive decoder, however small, is essential for producing a high quality final output from the encoder embedding.

In LibriSpeech, we report results comparing CTC, RNN-T, AED, and Aligners in Table 3. All models used the same 17-layer conformer encoder architecture and dimensions, except for differences in relative position attention, described in the following paragraph. Performance of our Aligner models was remarkably close to RNN-T, matched or beat AED, and clearly beat CTC. For each model, we report the best score from a small number of runs and checkpoints. Full training settings are provided in the appendix, along with a brief commentary on the relative performance between RNN-T and AED.

The distribution of training utterance lengths in LibriSpeech has a sharp drop-off after 17 seconds, and we observed both AED and Aligners losing performance when generalizing to longer test utterances, which run as long as 36 seconds. (See Tables 7,8 in the appendix for a breakdown by test utterance duration.) To improve performance of these models, we randomly concatenated a small subset of utterances in each train-

Table 3: WER (%) on LibriSpeech.

| | DEV | TEST-CLEAN | TEST-OTHER |
|---|---|---|---|
| CTC | 2.6 | 2.8 | 6.4 |
| RNN-T | 2.1 | 2.1 | 4.6 |
| AED | 2.2 | 2.4 | 5.5 |
| ALIGNER | 2.2 | 2.3 | 5.1 |

ing batch (*e.g.*, 15%), creating some examples up to 36 seconds. Thus, with adequate training, Aligners were able to self-transduce sequences of up to 900 frames. Learned relative attention encoding [46] trained slowly, so AED, CTC, and our model were trained with the faster Rotary Position Embedding [47] (RNN-T achieved slightly better test scores with learned relative encoding, reported in the table). In the following section, we discuss ways to use our model to perform long-form recognition to arbitrary lengths beyond what is seen in training.

## 4.4 Long-Form Recognition

It is not always practical to train with examples as long as the desired usage, and for any given Aligner-Encoder architecture there is likely some maximum alignable length. So it may be necessary to perform recognition on utterances longer than a trained model's capability. Our baseline method for comparing long-form recognition is "blind segmenting": 1) cut the audio into fixed-length, non-overlapping segments, 2) transcribe them separately, and 3) concatenate the resulting text segments. Perhaps the main shortcoming of blind segmenting is that recognition at the boundaries is prone to error, where the audio might cut in the middle of a word. One applicable solution is to use overlapping segments with an appropriate routine to stitch the hypothesis together in post-processing [48, 49]. Here, we instead describe how to improve continuity using the model itself, using only inference configurations which require no further training.

In our approach, cutting and re-concatenating the sequence happens within the model–called "chunking" to distinguish from the baseline. We preserve some continuity by cutting the audio only after the 2-D convolutional feature layers. The chunks are processed independently in parallel through the conformer, after which they are re-concatenated *before* decoding. The decoder processes the full-length embedding sequence into the full transcription hypothesis, using awareness of the chunk boundaries. Within each chunk, the decoder ignores frames after the first <EOS> emission, and it resumes decoding at the beginning of the next chunk. Ideally, the decoder's prediction network will carry its state forward across the chunk boundaries to evenly incorporate all tokens.

When we experimented with chunk sizes close to the training length, however, it led to increased deletions near the chunk ends. It was actually better to reset the prediction network state. This is understandable as the decoder was only ever trained to begin each utterance with a blank state, and it may be helping to count frames until <EOS>. The best performance, however, came from a middle approach: at the chunk boundary we reset the prediction network state but then "prime" it by re-processing the last several tokens through it. We found state-priming to reduce the number of errors at the boundary without raising later deletions.

For testing, we turn to the YouTube domain, where the training utterances ranged from 5-15 seconds long but test utterances spanned several minutes. Table 4 compares results from RNN-T and the Aligner-Encoder. They achieved equal WER (7.6%) with a 15-second blind segmenter, due to equivalent errors at the boundaries. In

Table 4: WER (%) on YouTube long-form test set.

|         | 15S SEGMENTED | UNSEGMENTED |
|---------|---------------|-------------|
| RNN-T   | 7.6           | 6.8         |
| ALIGNER | 7.6           | 7.3         |

the unsegmented arrangement, the RNN-T, with local attention of width 256, generalized well to the full utterance length; its WER improved to $6.8\%$[3]. Inside the Aligner, we applied a chunking period of 14 seconds (176 embedding frames). We found it best to reset the prediction network at the chunk boundaries and prime it with 10 tokens of history, which achieved 7.3% WER (7.5% without state-priming). While falling short of the RNN-T in this case, the Aligner did improve performance by smoothing the chunk boundaries, even without having been trained to do so. This result demonstrates our model's capability for long-form recognition, and could possibly be improved, such as by introducing additional training for this mode.

## 4.5 Alignments

### 4.5.1 Self-Attention Probabilities

To study how the encoder predicts the alignment, we examine the self-attention probabilities it computes during the forward pass. Figure 2 shows self-attention probabilities (every row sums to one) from the same head at different layers within one of our 17-layer encoders trained on LibriSpeech. A striking pattern emerges. One might expect the alignment to happen gradually in a network replete with residual connections, and indeed the self-attention in layer 4 is highly diagonal along the sequence; its output positions (vertical) draw from very near their own position from the input (horizontal). By layer 13, however, this pattern changes completely, and information concentrated in the front of the sequence appears to be distributed broadly. Suddenly, in the next two layers, the

---

[3]It is not automatically the case that RNN-T models exhibit good length generalization; it is by some combination of deliberate preparation and chance that our cases do.

audio-to-text alignment is clearly visible. The label contains 12 word-pieces, and precisely that many output positions receive information from points distributed monotonically along the inputs to these layers. The remaining output positions are possibly populated with filler values–these positions in the final output embedding receive no training. The alignment is complete even before the final layer, where an unrelated distribution is seen.

In this network, all the attention heads showed the same alignment operation happening in the 14th and 15th layers, for every input example we observed. Early to middle layers performed different mixtures of diagonal (local) and non-diagonal (global) attentions for different heads, which were less interpretable. Overall, the self-attention layers appear to be primarily performing audio encoding roughly in-place for many layers, and then they very explicitly perform the full alignment within as little as two layers. Interestingly, positions near the beginning and end of the sequence are perhaps used as margins for bookkeeping, where the training utterances often contain silence anyway.

The alignment itself is useful in many applications, and can be estimated from token emission positions when decoding with RNN-T. Our model has obscured that information, but the observed behavior affords the possibility of extracting alignments during inference. The bottom subplot of Figure 3 shows the average of all attention heads in Layer 15. Seen side by side, they closely track the RNN-T alignment, which is likely close to ground truth.

### 4.5.2 Embedding Alignment

To further study the alignment process, we also examined the intermediate embeddings themselves by the following procedure. Given the converged Aligner-Encoder model, we trained an RNN-T model of the same size, which is randomly initialized except that the parameter values from the beginning layers of the Aligner are used in the corresponding layers of the new encoder. The Aligner weights were kept frozen, so that the portion of the model trained by RNN-T receives as input the in-

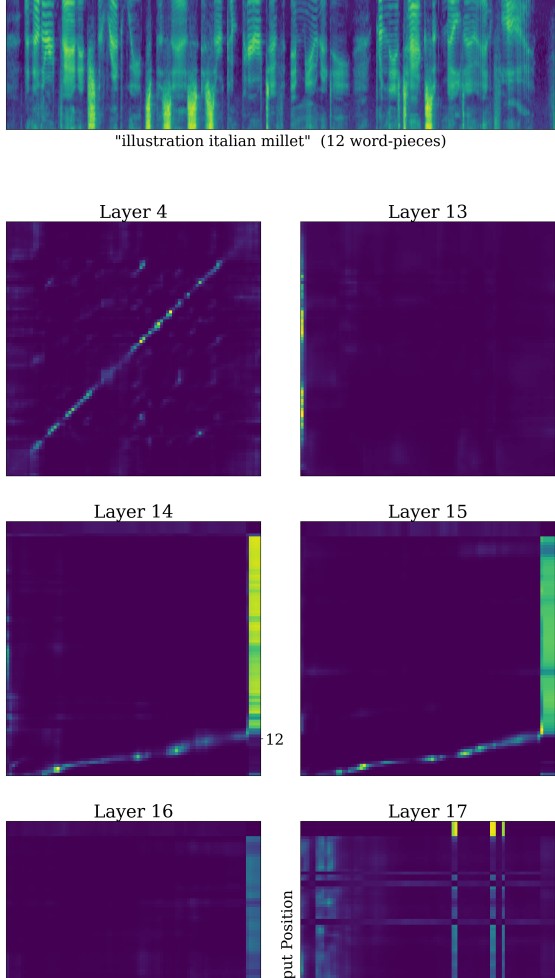

Figure 2: Self-attention probabilities from a single head at different layers in a 17-layer Aligner-Encoder performing audio-to-text alignment.

termediate Aligner embedding. Finally, we could measure alignments through the RNN-T decoder as usual–we computed the full forward-backward probabilities of the decoding lattice. By repeating this procedure using different numbers of Aligner-Encoder layers as the foundation, we traced the alignment progression of the internal representation through the forward pass of the original model.

The result is shown in Figure 3, where the suddenness of the process is apparent. Within a single layer the representation shifts from its original layout to become fully front-aligned with the output. This is the same layer that exhibited the most pronounced alignment self-attention probabilities previously. In the figure, each subplot is independently normalized, and the decoding lattice probabilities (computed from the forward and backward passes) are re-computed using a high temperature to exaggerate the tails of the distributions, which are tightly peaked. The bottom subplot shows the self-attention

probabilities averaged over all heads in Layer 15, normalized by row but not temperature-adjusted. The hot spots track closely with the original RNN-T alignment—note the lattice calculation fills in probability mass where the blank token is to be emitted, whereas the Aligner self-attention ignores those segments. The close correspondence of the pauses is especially visible.

### 4.5.3 Failure Mode Analysis

Using the same diagnostic technique, we can examine the failure mode of poor length generalization. Figure 4 shows such a case, for an utterance that is 1.5x longer than any training example. The model is only able to align a portion of the utterance, explaining why sometimes many deletions occur from dropping the latter part of the text. Interestingly, the RNN-T model trained atop the 15 frozen Aligner-Encoder layers (as in the previous subsection) still assigns some probability to the remainder of the label sequence, in a somewhat aligned fashion but pushed to the tail of the embedding sequence. This part is not decoded successfully. As we increased the number of Aligner layers used, we observed that the RNN-T hypotheses began to lose tail words several layers prior to the apparent alignment layer. Along with the self-attention visualization of Figure 2, this supports the conclusion that the Aligner works for several layers to prepare for the align-

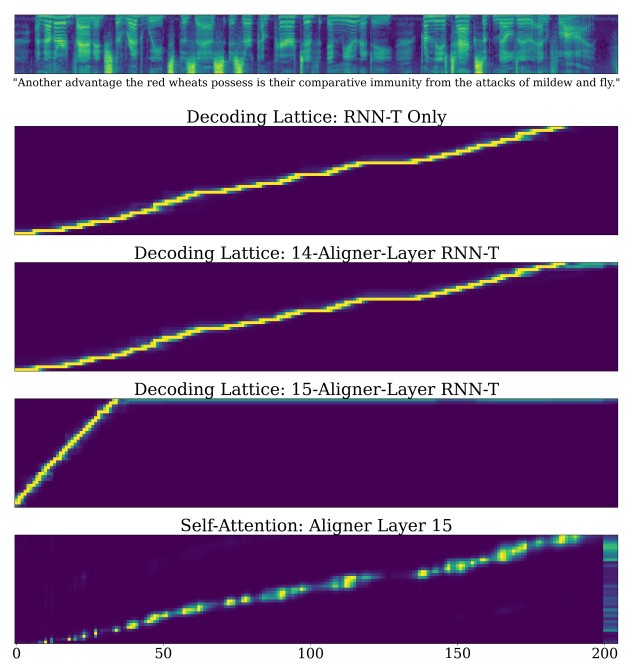

Figure 3: Decoding lattice probabilities ($U$ vs $T$) from RNN-T-on-Aligner and self-attention weights exhibiting successful alignment, within a specific layer.

ment operation; the over-length sequences already begin to be disrupted earlier. Separately, we note that for sequences which are near the length-capacity of the all-Aligner model, when deletions begin to occur, we observed them to often happen somewhere in the middle.

### 4.5.4 Reverse Alignment

We conducted one final experiment to demonstrate the likely applicability of Aligner-Encoders to tasks other than ASR. Specifically, we desired to show that our model has greater flexibility in aligning information than only the monotonic and *increasing* alignment relationship of standard ASR. To do so, we constructed the extreme case of *decreasing* alignment by completely reversing the audio input, so frames at the end of the audio input correspond to the beginning of the text output, and vice versa. Experimenting in LibriSpeech, the resulting model achieved similar performance as the regular model on utterances within the training length. Further, the self-

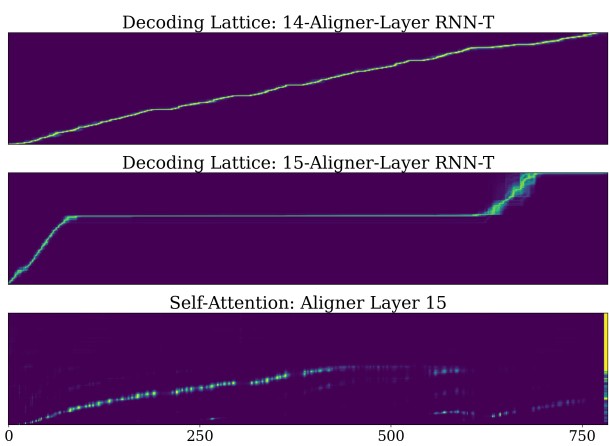

Figure 4: Decoding lattice probabilities ($U$ vs $T$) from RNN-T-on-Aligner and self-attention weights exhibiting a failure mode for an utterance 1.5x longer than trained.

attention weights exhibited the same behavior as in the forward model, except that in the aligning layer the weights trace from the upper-left to lower-right, showing the sequence reversal in action

(compare against the bottom panel of Figure 3). We include alignment plots for this model in the appendix, Figure 5. The equal ability to completely reverse the order of the information in the encoder embedding strongly suggests that our model could perform non-monotonic tasks, as well, which require more varied information re-ordering. Two prominent non-monotonic tasks where Aligner-Encoders have potential to simplify modeling include machine translation (MT) (*e.g.* [50]) and automatic speech translation (AST) (*e.g.* [51–54]).

### 4.6 Computational Efficiency

Considering training speed, memory usage, and inference speed, we observed favorable comparisons for our model owing to its reduced complexity. While the exact numbers will depend on many hardware and implementation details, we present a representative case using our 100M-parameter encoder LibriSpeech models in Table 5. Notably, the RNN-T spent roughly 290ms per training step in the decoder and loss (forward + backward propagation), whereas our model required only 29ms, a 10x gain. Our RNN-T loss implementation is highly optimized, so the majority of the gains came from eliminating the scan associated with the $T$ dimension of the joint network output. The peak memory usage for our model decreased by 18% (by 1.4G per accelerator)[4]. The 4-layer AED decoder trained with a step time essentially as fast as ours, thanks to good parallelization of transformers, and did tend to converge in significantly fewer steps (see A.2).

The inference speed of our model, however, was by far the fastest, especially relative to AED. We profiled all models using 11.5 seconds of audio, a batch size of 8, and beam search of size 6 (effective decoder batch size 48). The decode step for the Aligner and RNN-T[5] (LSTM + Joint network) measured a mere $190\mu$s. As discussed earlier, we can estimate the total decode time as the step time multiplied by the text length $(U)$ for our model, versus multiplying by the sum of the audio and text lengths $(T + U)$ for the RNN-T, re-

Table 5: Example measured compute times for our LibriSpeech models (lower is better).

| (MILLISECONDS) | AED | RNN-T | ALIGNER |
|---|---|---|---|
| TRAINING STEP: (ENCODER=560MS) | | | |
| DECODER+LOSS | 31 | 290 | 29 |
| TOTAL | 591 | 850 | 589 |
| INFERENCE: (ENCODE=32MS; T=300,U=100) | | | |
| DECODE STEP | 8.5 | 0.19 | 0.19 |
| DECODE | 850 | 76 | 19 |
| TOTAL | 832 | 108 | 51 |

sulting in substantial savings. We approximate with $T = 300, U = 100$. Turning to AED, each step through the transformer decoder was almost two orders of magnitude slower, at 8.5ms (and $U$ steps are executed).[6] The forward pass itself required 5ms per step, and re-ordering the decoder state as part of the beam search occupied the other 3.5ms. The re-ordering time was negligible in our model due to the minuscule LSTM state. Including the encoder, the total inference time of our model is estimated at 2x faster than RNN-T and 16x faster than AED in this scenario.

## 5 Conclusion

We have shown that transformer-based encoders are able to perform audio-to-text sequence alignment internally during a forward pass, using only self-attention (*i.e.*, prior to decoding, unlike previous models). This finding enables the use of a much simpler ASR model, the Aligner-Encoder, which has a simple training loss and achieves lower decoding complexity than past models. A key design point of our model is to keep the benefit of auto-regressive modeling while removing as much computation as possible from the auto-regressive loop. Desirable extensions to our model for ASR that would require further study include: use in streaming recognition, ability for multilingual recognition, and incorporation of traditional pretrained encoders, to name a few. Fusion with an external language model [24] could be simplified relative to RNN-T owing to the absence of blank tokens, and for the same reason our model may be more receptive to training on other losses such as from text-only data. Looking beyond ASR, application to machine (text) translation would require some modification to permit output sequences longer than the input, although speech translation would not. Altogether, Aligner-Encoders and their newly identified ability to learn self-transduction present promising opportunities for future research and application.

---

[4]Total training time was roughly a day and a half.

[5]We disabled path merging for RNN-T.

[6]Our implementation included both self-attention and cross-attention in all layers, and a decode cache.

## Acknowledgments and Disclosure of Funding

The authors would like to acknowledge the numerous members of the Google Speech Team who contributed either directly or indirectly through work on a shared code base, data preparation and maintenance, experiment and evaluation infrastructure, and furthermore by providing guidance and answering questions pertaining to these elements and other past research experience. We also thank the several anonymous reviewers who provided valuable feedback resulting in significant improvements in the quality of this paper.

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

# A  Appendix / supplemental material

## A.1  LibriSpeech – Expanded Settings and Results

Table 6: LibriSpeech common training settings.

| SETTING | VALUE |
|---|---|
| LEARNING RATE | 5.0 |
| LEARNING RATE DECAY | SQUARE-ROOT |
| LEARNING RATE WARMUP (STEPS) | LINEAR (10,000) |
| OPTIMIZER | ADAM |
| OPTIMIZER BETA-1, BETA-2 | 0.9, 0.98 |
| OPTIMIZER EMA DECAY | 0.9999 |
| CLIP GRAD NORM | 5.0 |
| L-2 REGULARIZER WEIGHT | 1E-6 |
| BATCH SIZE | 2,048 |
| TRAINING STEPS | 150,000 |
| VARIATIONAL NOISE SCALE | 0.075 |
| VARIATIONAL NOISE VARIABLES | TEXT EMBEDDING, LSTM |
| SPECTRUM AUGMENTATION | TIME & FREQ MASK |
| CONFORMER CONV 1-D KERNEL | 10 (RNN-T: 32) |
| SELF-ATTENTION HEADS | 8 |
| TEXT EMBEDDING SIZE | 128 (AED: 512) |
| LABEL SMOOTHING WEIGHT | 0.1 (RNN-T: N/A) |

Table 7: WER (%) on LibriSpeech Test-Clean set by utterance duration, including models trained with concatenated training examples covering up to the maximum test length of 36s. The number of test utterances in each category is 2466, 89, and 65, in order of length.

| TEST-CLEAN | < 17S | 17-21S | > 21S | ALL |
|---|---|---|---|---|
| CTC | 2.8 | 2.7 | 3.5 | 2.8 |
| RNN-T | 2.1 | 1.9 | 2.8 | 2.1 |
| AED | 2.3 | 2.1 | 15.3 | 3.3 |
| ALIGNER | 2.4 | 7.0 | 28.0 | 4.8 |
| AED-CONCAT | 2.4 | 2.1 | 2.8 | 2.4 |
| ALIGNER-CONCAT | 2.2 | 2.1 | 2.9 | 2.3 |

Table 8: WER (%) on LibriSpeech Test-Other set by utterance duration, including models trained with concatenated training examples covering up to the maximum test length of 36s. The number of test utterances in each category is 2834, 70, and 35, in order of length.

| TEST-OTHER | < 17S | 17-21S | > 21S | ALL |
|---|---|---|---|---|
| CTC | 6.6 | 6.0 | 4.3 | 6.4 |
| RNN-T | 4.7 | 4.4 | 3.1 | 4.6 |
| AED | 5.4 | 5.4 | 24.2 | 6.3 |
| ALIGNER | 5.2 | 8.4 | 29.2 | 6.5 |
| AED-CONCAT | 5.6 | 5.6 | 3.0 | 5.5 |
| ALIGNER-CONCAT | 5.2 | 5.3 | 3.3 | 5.1 |

## A.2  LibriSpeech – AED versus RNN-T Performance

While historically AED has out-performed RNN-T on LibriSpeech, this appears to have changed with the introduction of the Conformer [20]. To this point, there are several relevant factors worth noting for our case. Both models receive positional encoding following the feature convolution layers. Our RNN-T is trained and operated in a non-streaming mode (*e.g.*, with non-causal attention). We trained

AED with label smoothing (weight of 0.1) to prevent overfitting. Our AED models did often converge much faster, sometimes in as few as 25k training steps; we conducted a small hyperparameter search over learning rate and schedule to ensure against overfitting. Lastly, our AED decoder is a 4-layer transformer with 18M parameters (total model parameters 128M), which is already significantly larger than the 3.5M-parameter LSTM decoder for the RNN-T. A 148M-parameter conformer AED was reported in [55] to achieve $2.16\%$ and $4.74\%$ on Test-Clean and Test-Other, respectively, which is better than our smaller AED baseline but still less good than our RNN-T, which we keep as the relevant SOTA for comparison.

## A.3 Reverse Alignment Experiment Figure

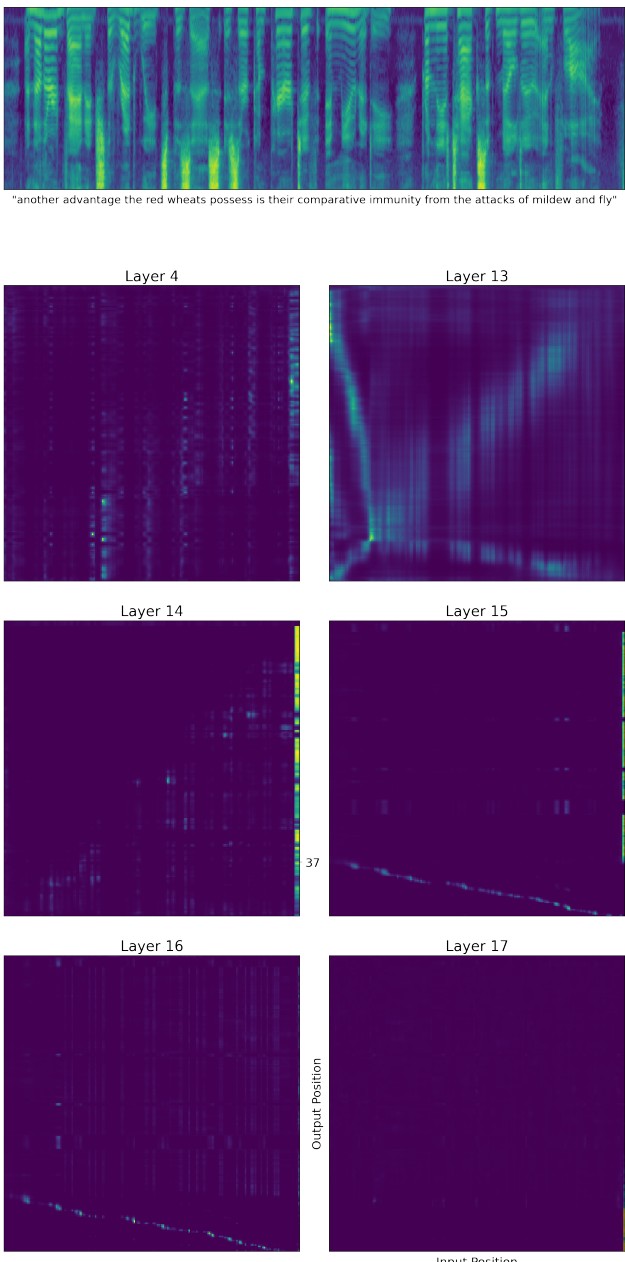

Figure 5: Self-attention weights in an Aligner-Encoder (from a single head) trained on reversed audio; the reverse alignment is clearly visible in layers 15 and 16. (LibriSpeech, 17-layer encoder).

