# OpenReview forum: "Aligner-Encoders: Self-Attention Transformers Can Be Self-Transducers"
_NeurIPS.cc/2024/Conference — NeurIPS 2024 spotlight_

### Official Review · Reviewer_VLfL · 2024-07-07

**Soundness:** 3
**Presentation:** 3
**Contribution:** 3
**Rating:** 7
**Confidence:** 5

**Summary:**

This paper proposes a new ASR model that connects a self-aligned encoder and the light text-only recurrence of RNN-T. The proposed model can be trained with label-wise cross entropy loss, which is computationally efficient than RNN-T training. The authors show the limitation of the model to inference on long-form audios and give a special inference configuration to mitigate the issue. Experiments on Librispeech and larger-scale ASR datasets demonstrated the close performance compared to other ASR models. The authors also show the audio-text alignment in the self-attention weights of a certain layer, which could be said to perform “self-transduction”.

**Strengths:**

The paper proposed a new ASR model for better training and inference efficiency. The idea can be similar to performing down-sampling on encoder side but is more aggressive (to token-level). The strengths of the paper are:
1. The proposed model consumes way less GPU memory than RNN-T and achieves much lower latency than AED.
2. The analysis of the text-audio alignment behavior in self-aligned encoder.
3. A modification on Aligner without training to enable long-form audio decoding.

**Weaknesses:**

The major concerns are the presentation and the limitation of the proposed model.
1. The explanation of the aligner modification for long-form audio is somewhat hard to follow. It may be better to have figures, or equations involved in the paragraph.
2. As the claim of the paper is that Aligner encoder achieves better efficiency than RNN-T and AED, it would be better to summarize the experimental numbers in Section 4.6 into a table for reader's convenience. People might be interested in those strong numbers.
3. The proposed model may have limited use cases e.g. offline ASR because the encoder cannot be implemented for streaming with the current design.
4. A related work that replaces RNN-T loss with the cross-entropy loss and reduces the GPU memory usage. (https://arxiv.org/pdf/2307.14132).

**Questions:**

Several additional questions:

1. If I remember correctly, the AED model can achieve no worse WER than RNN-T on Librispeech in literature, I am wondering did the authors look into other reasons why AED in Table 3 is much worse than RNN-T (except the long-form problem)?
2. Any results comparing rotary positional embedding and relative positional embedding for speech encoder on long-form audio? The authors mentioned that RNN-T used RPE, is it the reason why RNN-T is good at long-form audio?

**Limitations:**

The main limitation is the use case of the proposed method, aka, non-streaming applications. Even though, the method can be helpful for many different purposes.

---

> ### Author Rebuttal · Authors · 2024-08-06
>
> Thank you for the close review of our work.
>
> Thank you for the suggestion to revise the section on the long-form inference modification.  In earlier drafts of the paper, this has been the most difficult part to write clearly.  Space permitting, a figure or perhaps a pseudo-algorithm would be helpful, we will continue to revise to try to improve this section.
>
> Good idea to highlight the computational efficiency / latency results in a table.
>
> It is true we make no claims toward streaming capability.  It is possible that our model is capable of performing streaming recognition like RNN-T models trained for this purpose (although perhaps not with as low of latency).  Or it might require adaptations similar to what has been done to make Whisper streaming-capable, for example: https://aclanthology.org/2023.ijcnlp-demo.3.pdf.  Since our model decodes non-streaming with much lower latency than AED (owing to our small decoder), it is likely that such adaptations could produce a much lower latency in our model.  It would definitely be worthwhile to investigate streaming capabilities of models based on our Aligner Encoder.  Since this potential limitation is not only for our model, but also applies to AED, and it could require significant further study and reporting, we hope that it does not diminish too much the significance/relevance of our submission, where we have included many other detailed analyses and ablations.
>
> Yes, we have cited https://arxiv.org/pdf/2307.14132 in our manuscript: [34].
>
> Another reviewer also asked about why AED could be worse than RNN-T on LibriSpeech.  We believe this reversal happened with the introduction of the Conformer, as that paper shows RNN-T achieving better performance than transformer-AED.  In researching this question we also discovered a reference that reports a conformer-AED on LibriSpeech (https://arxiv.org/abs/2210.00077), which is still not quite as good as RNN-T (so the SOTA we are comparing against remains the same).  It is better than our conformer-AED result, but it also used more learnable parameters.  Still, we should include it as a point of comparison and double-check all the settings.  We welcome any other suggested references or explanations.
>
> Good question about RoPE versus relative position encoding.  In the RNN-T baselines we ran, the training run with relative positional embedding produced very slightly better results than with RoPE (e.g. 2.1 versus 2.2, 4.6 versus 4.7), so we reported the relative positional embedding results.  However relative positional embedding is significantly slower to train than RoPE, so for the remaining LibriSpeech models (including ours) we used RoPE.

---

> > ### Comment · Reviewer_VLfL · 2024-08-10
> > **Reply to the rebuttal**
> >
> > Thanks for the rebuttal. The paper can benefit from the revisions proposed by the authors. I am willing to raise the score to 7 (Accept).
> >
> > In terms of the AED vs. RNN-T, it would be great to clarify the settings and explain the gap. My impression is that RNN-T typically uses a streaming encoder for online purpose, while AED uses non-streaming decoder for offline ASR. AED would always be better in terms of WER. Maybe the authors used the same decoder, and I didn't check the paper again. The authors can use several sentences to clarify the gaps in the revised version of the paper.

---

> > > ### Author Response · Authors · 2024-08-12
> > > **AED vs RNN-T**
> > >
> > > Thank you!  Indeed the AED vs RNN-T performance seems to be a valuable question, we're not sure we'll be able to answer it definitively, but will add more details and some discussion (please also see comments under reviewer JgvT).  In our experiments we used the same encoder for both, including global attention--so no streaming, which as you mentioned might otherwise put RNN-T at a disadvantage.  It might simply be that AED requires a larger decoder to do really well, where we only used a 4-layer, 18M parameter transformer (already much larger than the LSTM used for RNN-T, 3.5M parameters).

---

### Official Review · Reviewer_Btkf · 2024-07-09

**Soundness:** 3
**Presentation:** 3
**Contribution:** 3
**Rating:** 7
**Confidence:** 2

**Summary:**

The paper introduces Aligner, which is to take the best parts from RNN-Transducer and AED (Attention Encoder-Decoder) models. The idea comes from the intuition that the transformer encoder with self-attention can already learns to align the input and the output -- which is explicitly modeled by previous approaches like applying techniques to ensure the monotonic alignment or doing dynamic programming to find the alignment. Aligner just simply train with cross-entropy loss, and the results surprisingly show that the encoder can internally learn to align the input and the output during the forward pass. Experimental results showed that Aligner can perform comparably with existing baselines, and show the computational efficiencies.

**Strengths:**

- The motivation about the computational efficiencies of existing models and the idea of developing Aligner is very clear.
- The idea is derived from the observation about internal alignment of Transformers, which give meaningful insights to the readers.
- It successfully tackles the limitations of existing ASR models with simple and intuitive ways, and the results show that it works well.
- The paper includes sufficient amount of evaluation results and showing the limitations (in 4.5.3) as well, providing insights to the readers about these models much.
- I feel the writing is also very clear.

**Weaknesses:**

- The idea is yet confirmed in the model fine-tuning setups; as recent Whisper model [1] shows that large-scale pretraining can leads to performant ASR models, I hope this Aligner work applied to the large-scale pretraining setups and show the effectiveness as well. Note that I feel this is not a critical weakness of this paper.
- Honestly, I am not following the most recent state-of-the-art models for ASR systems, and not sure if there are some missing baselines that the authors should also compare with. I am willing to listen to other reviewers' opinions about this.

[1] Robust Speech Recognition via Large-Scale Weak Supervision, Radford et al, 2022

**Questions:**

No questions for now.

**Limitations:**

Limitations are adequately addressed.

---

> ### Author Rebuttal · Authors · 2024-08-06
>
> Thank you for considering our work closely, your summary is accurate to what we intended.
>
> It is a worthwhile question whether pre-training can be combined with our model.  As it stands, our model seems to use several of the same layers in the conformer for 1) encoding and 2) alignment, whereas existing pre-training methods will only train encoding.  It would be interesting for example to take an existing good encoder (e.g. from RNN-T) and see if it can be fine-tuned for a small number of steps to learn the alignment, rather than needing to start from scratch.  We may attempt this experiment, thank you for the suggestion.

---

> > ### Comment · Reviewer_Btkf · 2024-08-12
> >
> > Thank you, and I will keep my rating as the score is already 7.

---

### Official Review · Reviewer_RgJ5 · 2024-07-10

**Soundness:** 4
**Presentation:** 3
**Contribution:** 4
**Rating:** 7
**Confidence:** 5

**Summary:**

This paper proposes a new speech recognition (or, more generally, sequence-to-sequence) architecture. This architecture performs an alignment process between input and output features via self-attention mechanisms in an encoder. The decoder network is a simplified version of the combination of the RNN-T or AED decoder, and its relationship is detailed in Section 2.2. The primary advantage of this method is reducing the computational cost by avoiding the dynamic programming to adjust the input and output length in RNN-T or cross-attention to consider all possible scores across the input frame and output token. The experiments show the comparable performance of the proposed method to RNN-T and AED, but it significantly reduces the computational costs.

**Strengths:**

- Novel speech recognition architecture or, more generally, novel sequence-to-sequence architecture.
- Comparable performance to other SOTA speech recognition architecture (AED and RNN-T) while reducing the computational complexity.
- Interesting analysis of the alignment behaviors and detailed ablation studies

**Weaknesses:**

- Weak reproducibility due to the use of non-public data for main ASR experiments and the lack of source code release. Note that I did not penalize this point in my initial judgment. But if this part is improved, I'll raise my score.

**Questions:**

- Section 2.1: Do you need $U \leq T$? Can we apply this method for $T \leq U$? This would happen in the general sequence-to-sequence problem like MT or TTS.
- Section 4.2: Did you try it with the other encoder layer than the Conformer (e.g., vanilla transformer)? I'm curious because the alignment properties of this method might depend on the convolution operation in the Conformer.
- Section 4.5.2: I'm not sure how each attention head behaves. Can you discuss a bit more about how this behavior is different across the head?
- Section 4.5: Do you have some results on MT or AST?

Suggestions
- I recommend the authors emphasize the practical benefit of the computational complexity of this method in the abstract.
- Sections 4.5.3 and 4.5.5: These sections provide the main experimental results in the appendix, which are not recommended. This way, it breaks the page limit rule and is unfair compared with the other papers, which put all main results in the main body. These sections should be rewritten to avoid using the appendix results. Note that some supplemental use of the appendix is no problem (e.g., Table 5 is a good example. This is too detailed and may not be so crucial for understanding the main idea of this paper, but it is essential for reproducibility. So, it is adequate to be located in an appendix section.).

**Limitations:**

- Since this is not based on a hard alignment approach (RNN-T and CTC are based on hard alignment), I'm curious about the hallucination issues often observed in AED or decoder-only architectures. For example, OpenAI's Whisper is based on AED (soft alignment). It has a serious hallucination issue (despite its outstanding performance), and I think this is a potential limitation of the soft alignment-based approaches in general. I want the authors to discuss this aspect. This method probably has an advantage over AED due to its shallow decoder architecture, but I'm not very sure.

---

> ### Author Rebuttal · Authors · 2024-08-06
>
> Thank you for your careful consideration of our work.
>
> Important question about the requirement for U <= T, especially for machine-translation.  One possible solution to extend to U>T would be to pad the input with some (fixed) number, P, of learnable input frames, which would provide the model the ability to write T+P output tokens.  (This is similar to the "registers" for vision transformers: https://arxiv.org/abs/2309.16588.)  Another possibility, for U >> T, would be to train the model to decode two (or more) tokens per embedding frame.   We did not need these changes for any of our ASR experiments, but would like to include them as suggestions in a revised discussion section.
>
> We only tried Conformer because it tends to perform better and was already in use for many pre-existing baselines, so we could expect the encoder quality to degrade without the convolutions (even if the alignment is still possible to perform solely with the transformer).
>
> In Figure 2, we plotted attention weights for a single head.  The patterns do look different for different heads in the early layers (this also happens in RNN-T), but we found them all to express the alignment as seen in Layers 14 & 15.  So in Figure 3 we were able to average across all heads when showing the alignment in Layer 15, which is probably more accurate than relying on a single head.
>
> Unfortunately we do not have expertise or infrastructure for MT or AST---we welcome any correspondence from future readers over future work!
>
> Regarding sections 4.5.3 and 4.5.4 figures, we placed images for these ablations into the appendix so they could be printed as large as possible.  On re-reading, we did not adequately describe the figure from 4.5.4 for the text to serve as a standalone result--will revise!
>
> On hallucinations, after looking at many, many examples, we did not observe this issue in any of our models, and did not see abnormally high insertion errors.  In other work, we have observed hallucinations when using a larger language model in AED-style, so it seems this is a characteristic of LLMs. It's possible that we have some small number of tokens hallucinated during silence only, which RNN-T models may also do.
>
> It is unfortunate that we are unable to release some of the datasets and our actual code--our hope is that the result on LibriSpeech (public) is sufficient for reproduction, paired with the fact that our method is only a simplification over previous models.  We are happy to correspond with anyone re-implementing in a public code repository.

---

> > ### Comment · Reviewer_RgJ5 · 2024-08-08
> > **Thanks for your explanation**
> >
> > The answers are valuable (especially for the head and hallucination discussions), but they are not related to the overall discussions. I already gave the accept score and I want to maintain it.
> >
> > I'm looking forward to MT or AST experiments and the open-source implementation of this method.

---

### Official Review · Reviewer_JgvT · 2024-07-12

**Soundness:** 3
**Presentation:** 3
**Contribution:** 3
**Rating:** 7
**Confidence:** 5

**Summary:**

A new simplified encoder-decoder model is presented without the attention. The decoder is generating the labels auto-regressively as usual until end-of-sentence (EOS). In contrast to attention-based encoder-decoder (AED) models, the attention is replaced by simply taking the same frame in the encoder - i.e. in decoder step u, it will take frame u also from the encoder. Thus the output sequence can never be longer than the input sequence. The idea is that the encoder with self-attention can already realign the information as necessary to output it label by label. The remaining encoder output frames after EOS are ignored, but of course all intermediate encoder frames are used due to self-attention.

It's an interesting test to see whether the self-attention is enough to already learn this. And the answer is yes, it can learn this.

Experiments are performed on three speech recognition tasks:

* Librispeech with 960h train data
* Voice Search with 500kh train data
* YouTube videos with 670kh train data

In all cases, a Conformer encoder is used. Word pieces are used as output labels.


The self-attention weights are analyzed and it is observed that the realignment happens in layer 14. This is also verified in another way, by freezing the first N layers of the encoder, randomly resetting the other encoder parameters, and adding a RNNT on top of it and training that, and then generating the RNNT soft alignment. With N>14, the alignment looks like the identity mapping.

**Strengths:**

Interesting idea and model.

The work shows that this simple idea seems to work, even though its performance stays behind the other existing models (CTC/RNNT/AED).

Interesting analysis on the attention weights and retraining the encoder partly and looking at the RNNT soft alignment.

**Weaknesses:**

No source code to reproduce the results?

No references are given to Voice Search and YouTube data, so it's impossible to reproduce and verify the results.

**Questions:**

How is the convergence rate in comparison to CTC, RNNT, AED? How is the alignment behavior early in training?

Table 3, what is dev, is that dev-clean, dev-other, or both combined?

Table 3, it seems a bit weird to me that the RNN-T is so much better than the AED model. I would actually expect the opposite. For example:

* A Comparison of Sequence-to-Sequence Models for Speech Recognition,
https://www.isca-archive.org/interspeech_2017/prabhavalkar17_interspeech.html
* On the Comparison of Popular End-to-End Models for Large Scale Speech
Recognition, https://www.microsoft.com/en-us/research/uploads/prod/2020/11/template-5fa34dc776e7f.pdf
Both show that AED is better than RNN-T. And this is what I have seen on many other occasions as well. Was this expected to you? Why? Or if not, how do you explain it?

What happens when f_pred is just a feed-forward network without the recurrence (dependence on g_{i-1}), i.e. you would get a model with only the last label as context? For RNN-T, it has been shown that this performs equally well as using the whole history. It would be interesting to see how it behaves for this model. (For an AED model, this is not really possible because the cross attention mechanism needs it.)

**Limitations:**

-

---

> ### Author Rebuttal · Authors · 2024-08-06
>
> Thank you for reviewing our work closely.
>
> The convergence rates are similar among CTC, RNN-T, and our model (good checkpoints are between 100k-150k training steps on LibriSpeech).  Interestingly, AED models sometimes did converge faster (as fast as 25k training steps for the best checkpoint).  We did not investigate this further other than to run AED again with a lower learning rate in case that was too high, but we weren't able to find a better result.  So in this regard our model is more similar to RNN-T.
>
> In Table 3 Dev is Dev-Clean only, we will edit the column label.
>
> This is an interesting question about AED versus RNN-T.  A possible explanation is the use of the conformer encoder.  "A Comparison of Sequence-to-Sequence Models for Speech Recognition" uses attention architectures that predate transformers.  "On the Comparison of Popular End-to-End Models for Large Scale Speech Recognition" finds an improvement using transformer-AED over RNN-AED, but it only reports RNN-T in its Table 1.  Our results more closely match those of the Transformer-AED and Conformer-Transducer given in the Conformer paper https://arxiv.org/abs/2005.08100 (Table 2) which also experiments on LibriSpeech.  In searching around for this question, we found a paper we could cite which gives a score for Conformer-AED on LibriSpeech is: https://arxiv.org/abs/2210.00077, which reports test-clean 2.16% and test-other 4.74%.  This is better than our AED results (with conformer), although they use more learnable parameters: 148M versus ours at 128M. It is still less good than our RNN-T (conformer), so it would not change the SOTA against which we are comparing.  Still, it seems worth citing and including in the comparison, and investigating their other settings.
>
> This is a very interesting question about running our model with limited recurrence in the decoder.  It is true that RNN-T models can often perform well using a history of only the 2 most recent tokens.  Time permitting we will attempt to launch this experiment, thank you for the suggestion.
>
> Unfortunately our implementation cannot be disentangled from a code base which cannot be shared.  We hope that the fact that our model is a simplification over previous models, and uses more standard deep learning components, will make it relatively easier to reproduce the results, and we have attempted to include extensive hyperparameters.

---

> > ### Comment · Reviewer_JgvT · 2024-08-08
> >
> > Thanks for the rebuttal.
> >
> > On AED vs RNN-T: Do you have exactly the same hyper-parameters for the encoder in both cases? Or were they tuned individually?
> >
> > I guess in the original Conformer paper, both the architecture and its hyper-parameters were optimized always using the same RNN-T sequence modelling on top. So this gives maybe an advantage for RNN-T over AED.
> >
> > RNN-T and CTC don't need any further positional information in the encoder output, while AED needs some information for the cross attention to work properly, such that it can know where to attend next. The architecture can indirectly learn absolute positional information, e.g. via the convolutional padding, but you can imagine that this is maybe not optimal. So maybe a different frontend, or explicitly adding absolute positional encoding would greatly help the AED sequence model.
> >
> > Just some thoughts on this. My intuition tells me that AED is still more powerful than RNN-T when this is taken into account. And/or when the architecture and/or frontend is tuned for AED.
> >
> > But studying this is probably out-of-scope for this work. Maybe the only reasonable simple experiment you could do now is adding absolute positional encoding to the encoder.

---

> > > ### Author Response · Authors · 2024-08-12
> > > **AED vs RNNT**
> > >
> > > Good questions, on AED versus RNN-T, we used the same encoder architecture for each.  In fact they both include an absolute positional encoding, added into the embedding after the initial 2-D convolution layers, prior to the first conformer layer (we need to add this to Table 5)--some early ablations showed this might not be critical but I think we didn't try again with all the other final settings to be sure.  One possible difference is that the variational noise is applied to the LSTM and the text embedding variable in RNN-T, and this is helpful to get a last bit of performance improvement, whereas in AED we only applied it to the embedding variable, since the decoder is much bigger.  Another difference is that in AED we needed label-smoothing, but this doesn't apply in RNN-T.  In both models we gave the encoder global attention, so it can operate on the whole sequence (no streaming).  It is a bit strange that for our AED to work on the longest test utterances, we needed to concatenate training examples (as we described), and we think it is generally known that AED struggles to generalize to longer lengths, but the other references we've found don't mention this.  Separately from this, it's possible that to perform better AED simply needs a larger decoder, since we used only a 4-layer transformer, which already adds 18M parameters.
> > >
> > > It's interesting that multiple reviewers have raised this question--seems worthwhile for us to add a short discussion about this.  Thank you.

---

### Comment · Area_Chair_3QCK · 2024-08-08
**Discussion period starts**

Dear Reviewers,

The authors have already posted their rebuttal to address review concerns. It is now the discussion period (Aug. 7 - 13). Please read the author response, and conduct the discussion. Thank you!

Thanks,
AC

---

### Decision · Program_Chairs · 2024-09-25

**Decision:**

Accept (spotlight)

**Comment:**

This paper introduces the Aligner, a simple yet efficient end-to-end (E2E) model that merges the benefits of RNN-Transducer (RNN-T) and Attention-based encoder-decoder (AED) for automatic speech recognition (ASR). The authors found that the transformer-based encoder is sufficient to handle alignment between speech and text. Leveraging this finding, they utilized the proposed aligner encoder to reposition relevant information to the start of the embedding sequence, enabling them to employ frame-wise cross-entropy loss from AED during training instead of relying on the intricate dynamic programming required by RNN-T, thereby simplifying the training process significantly. During inference, the Aligner demonstrates reduced decoder complexity compared to AED. The new model achieves recognition accuracy similar to that of popular AED or RNN-T models.

This paper presents a novel approach that is poised to make a significant impact on the speech recognition community. AED and RNN models currently dominate E2E models in ASR, and although numerous new architectures have been proposed recently, none has achieved the success of the Aligner introduced in this paper. It is impressive how this new model seamlessly integrates the strengths of AED and RNN-T models. The efficacy of the proposed method has been demonstrated across three distinct tasks: Librispeech from the public domain, an in-house voice search, and YouTube test sets, thereby reinforcing the credibility of the results.

The reviewers unanimously agreed this is a high-quality paper and recommended it for acceptance. In the rebuttal, the authors addressed most of the reviewers' questions. Several reviewers questioned why RNN-T outperforms AED in this study, given that AED is generally seen as more powerful while RNN-T is better for streaming. The authors explained that both models have an identical speech encoder setup and cited the Conformer-Transducer paper to support their argument. However, Table 1 in "A Comparison of Sequence-to-Sequence Models for Speech Recognition" shows RNN-AED outperforming RNN-T. I believe the performance difference between AED and RNN-T architectures should be less dependent on the speech encoder, whether it's a Transformer or LSTM.

In brief, this is an excellent paper featuring a new E2E architecture that merges the strengths of AED and RNN-T. It has the potential to significantly influence the speech community.